# Identifying bedrest using waist-worn triaxial accelerometers in preschool children

J. Dustin Tracy[ID]¹, Thomas Donnelly², Evan C. Sommer³, William J. Heerman³, Shari L. Barkin³, Maciej S. Buchowski[ID]²*

1 Economic Science Institute, Chapman University, Orange, California, United States of America, 2 Division of Gastroenterology, Hepatology and Nutrition, Department of Medicine, Energy Balance Laboratory, Vanderbilt University Medical Center, Nashville, Tennessee, United States of America, 3 Department of Pediatrics, Vanderbilt University Medical Center, Nashville, Tennessee, United States of America

* maciej.buchowski@vanderbilt.edu

## Abstract

### Purpose

To adapt and validate a previously developed decision tree for youth to identify bedrest for use in preschool children.

### Methods

Parents of healthy preschool (3-6-year-old) children (n = 610; 294 males) were asked to help them to wear an accelerometer for 7 to 10 days and 24 hours/day on their waist. Children with ≥3 nights of valid recordings were randomly allocated to the development (n = 200) and validation (n = 200) groups. Wear periods from accelerometer recordings were identified minute-by-minute as *bedrest* or *wake* using visual identification by two independent raters. To automate visual identification, chosen decision tree (DT) parameters (*block length*, *threshold*, *bedrest-start trigger*, and *bedrest-end trigger*) were optimized in the development group using a Nelder-Mead simplex optimization method, which maximized the accuracy of DT-identified bedrest in 1-min epochs against synchronized visually identified bedrest (n = 4,730,734). DT's performance with optimized parameters was compared with the visual identification, commonly used Sadeh's sleep detection algorithm, DT for youth (10-18-years-old), and parental survey of sleep duration in the validation group.

### Results

On average, children wore an accelerometer for 8.3 days and 20.8 hours/day. Comparing the DT-identified bedrest with visual identification in the validation group yielded sensitivity = 0.941, specificity = 0.974, and accuracy = 0.956. The optimal *block length* was 36 min, the *threshold* 230 counts/min, the *bedrest-start trigger* 305 counts/min, and the *bedrest-end trigger* 1,129 counts/min. In the validation group, DT identified *bedrest* with greater accuracy than Sadeh's algorithm (0.956 and 0.902) and DT for youth (0.956 and 0.861) (both P<0.001). Both DT (564±77 min/day) and Sadeh's algorithm (604±80 min/day) identified significantly less bedrest/sleep than parental survey (650±81 min/day) (both P<0.001).

**Data Availability Statement:** All raw data files are available from the Figshare database: https://figshare.com/articles/dataset/Train_Files/12904124

and https://figshare.com/articles/dataset/Test_Files/12904127.

**Funding:** This study was supported by the National Institutes of Health (NIH) grants HL103620 and HL103561 from the National Heart, Lung, and Blood Institute and the Eunice Kennedy Shriver National Institute of Child Health and Development, and the Office of Behavioral and Social Sciences Research: https://nam04.safelinks.protection.outlook.com/?url=https%3A%2F%2Fwww.nih.gov%2F&data=04%7C01%7Cmaciej.buchowski%40vanderbilt.edu%7C876b97453d1f47b15de808d8b8a53769%7Cba5a7f39e3be4ab3b45067fa80faecad%7C0%7C0%7C637462367312593967%7CUnknown%7CTWFpbGZsb3d8eyJWIjoiMC4wLjAwMDAiLCJQIjoiV2luMzIiLCJBTiI6Ik1haWwiLCJXVCI6Mn0%3D%7C1000&sdata=PnXgEOgTcQyoZQ8LVFVd1rAn9y0Y6KL3YnIYBw3yyw%3D&reserved=0. Additional funds were provided by grants DK058404 and DK20593 from the National Institute of Diabetes and Digestive and Kidney Diseases: https://nam04.safelinks.protection.outlook.com/?url=https%3A%2F%2Fwww.nih.gov%2F&data=04%7C01%7Cmaciej.buchowski%40vanderbilt.edu%7C876b97453d1f47b15de808d8b8a53769%7Cba5a7f39e3be4ab3b45067fa80faecad%7C0%7C0%7C637462367312593967%7CUnknown%7CTWFpbGZsb3d8eyJWIjoiMC4wLjAwMDAiLCJQIjoiV2luMzIiLCJBTiI6Ik1haWwiLCJXVCI6Mn0%3D%7C1000&sdata=PnXgEOgTcQyoZQ8eLVFVd1rAn9y0Y6KL3YnIYBw3yyw%3D&reserved=0. The REDCap Database was supported by TR000445 from the National Center for Advancing Translational Sciences: https://nam04.safelinks.protection.outlook.com/?url=https%3A%2F%2Fwww.nih.gov%2F&data=04%7C01%7Cmaciej.buchowski%40vanderbilt.edu%7C876b97453d1f47b15de808d8b8a53769%7Cba5a7f39e3be4ab3b45067fa80faecad%7C0%7C0%7C637462367312593967%7CUnknown%7CTWFpbGZsb3d8eyJWIjoiMC4wLjAwMDAiLCJQIjoiV2luMzIiLCJBTiI6Ik1haWwiLCJXVCI6Mn0%3D%7C1000&sdata=PnXgEOgTcQyoZQ8eLVFVd1rAn9y0Y6KL3YnIYBw3yyw%3D&reserved=0. WJH was supported by K23 HL127104 grant from the National Heart Lung and Blood Institute: https://nam04.safelinks.protection.outlook.com/?url=https%3A%2F%2Fwww.nih.gov%2F&data=04%7C01%7Cmaciej.buchowski%40vanderbilt.edu%7C876b97453d1f47b15de808d8b8a53769%7Cba5a7f39e3be4ab3b45067fa80faecad%7C0%7C0%7C637462367312593967%7CUnknown%7CTWFpbGZsb3d8eyJWIjoiMC4wLjAwMDAiLCJQIjoiV2luMzIiLCJBTiI6Ik1haWwiLCJXVCI6Mn0%3D%7C1000&sdata=PnXgEOgTcQyoZQ8eLVFVd1rAn9y0Y6KL3YnIYBw3yyw%3D&reserved=0. The funders had no role in study design, data collection

## Conclusions

The DT-based algorithm initially developed for youth was adapted for preschool children to identify time spent in *bedrest* with high accuracy. The DT is available as a package for the R open-source software environment ("PhysActBedRest").

## Introduction

Sleep is fundamental for optimal children's health and wellbeing [1–3]. Low sleep quality in preschool children is associated with delayed growth, a decline in daytime physical activity, body weight abnormalities, and learning and behavioral difficulties [4,5]. Young children require increased sleep duration to ensure healthy growth and development [6], and yet, there are no available methods to measure sleep reliably and conveniently in natural environments. The current clinical gold standard for assessing sleep in children is polysomnography (PSG), which uses advanced laboratory-based technology to assess brain, cardiovascular, and respiratory activity during nighttime sleep [7–9]. PSG yields several sleep metrics, including sleep onset and offset sleep stages. Some sleep metrics can be approximated through other means, including accelerometers [10].

Recently, accelerometry, also known as actigraphy, has been used to assess habitual sleep-wake patterns in children, both healthy and with sleep disorders [11–13], using relatively non-intrusive research-grade triaxial accelerometers worn for several days or weeks [14]. Although accelerometry does not provide detailed information on sleep that PSG does, it has the advantages of portability, tolerability, and the ability to identify and differentiate between nighttime and daytime sleep and non-sleep sedentary behaviors and physical activity measured for several days or weeks [15]. The primary signal used by accelerometers to identify sleep is a lack of body movement confounded by slight movements during sleep and lack of body movement during wake episodes [14,16]. The results may not necessarily result in the same sleep period (s) s identified by PSG or a parental report.

Field-based sleep assessments in preschool children also frequently rely on parent-report methodologies, often not providing sufficiently valid or reliable sleep estimates [12,17,18]. Combining PSG, accelerometry, and parental or self-reports could help balance the limitations of each. Because this is often not feasible in both research and clinical studies, accelerometry is frequently used as a method of choice [19].

In children, accelerometry is 80 to 90% concordant with PSG during night rest [20–22]. A significant challenge in preschool children is to capture time spent in nighttime sleep and daytime naps over several days in a natural environment, without additional information from a parental report [23]. The majority of automated algorithms for assessing sleep were validated in 10-18-yr-old children [24], no. Belanger et al. [9] developed two algorithms for the wrist-worn and ankle-worn single-axis Actiwatch accelerometer using data from 2–5 years old children (n = 12) during nighttime sleep. However, no triaxial accelerometer's-based algorithm validation studies in healthy preschool children have been reported to our knowledge. The present study aims to fill a gap in the literature by identifying periods of bedrest that include nighttime sleep and daytime naps with a high degree of accuracy using accelerometer recordings collected for several days.

We previously developed and implemented as an R software program a decision tree (DT) that uses an automated series of decision rules with user-modifiable parameters to identify the time in bedrest in healthy 10-18-year-old children (DT-Youth) with good accuracy [25].

and analysis, decision to publish, or preparation of the manuscript. There was no additional external funding received for this study.

**Competing interests:** The authors have declared that no competing interests exist.

However, its validity cannot be assumed for younger children with different personal characteristics, physical activity levels, and sleep habits.

The goal of this study was to adapt DT-Youth to identify bedrest periods in preschool children using a large group of 3-6-year-old children living in their homes and wearing waist-worn triaxial accelerometers. The adapted DT algorithm's performance was compared to that of Sadeh's automated sleep scoring algorithm, which is frequently used in pediatric research for sleep assessment from accelerometer recordings [26], to previously validated DT-Youth, and to the parental survey of child sleep duration. We posited that the new adapted DT would better identify bedrest in preschool children than Sadeh's algorithm, DT-Youth, and parental survey.

## Methods

### Participants

The study included a subset of 610 healthy 3-6-yr-old children, 294 males, BMI percentile ≥50th and <95th recruited to The Growing Right Onto Wellness (GROW), a 36-month behavior change intervention trial focused on childhood obesity prevention among preschool-aged children from underserved communities in Nashville, Tennessee and registered with Clinical-Trials.gov (NCT03268577) [27]. All applicable institutional and governmental regulations concerning human volunteers' ethical use followed the Helsinki-II Declaration's ethical principles. The Institutional Review Board of Vanderbilt University approved the study protocol and consent form (IRB No. 120643). Written informed consent was obtained before the study. Study data were collected from 2012 to 2017, and current analyses were performed in 2019 and 2020.

### Anthropometric and demographic data

Standing height (to the nearest 0.1 cm) and weight (to the nearest 0.1 kg) were measured using wall-mounted stadiometers and research-grade scales using a rigorous protocol [28]. Body mass index (BMI) was calculated as a ratio of weight (kg) and height ($m^2$). BMI percentiles were calculated using the Centers for Disease Control (CDC) age- and sex-specific growth references. Chronological age was determined from birth dates and the accelerometer wearing start date.

### Accelerometry recordings

Parents were asked to help their children wear an ActiGraph GT3X+ triaxial accelerometer (ActiGraph, Pensacola, FL) on the waist for at least 7 days, including sleeping and napping. Data were collected at baseline and after 12, 24, and 36 months of the trial. The accelerometers collected recordings at 40 Hz chosen at the beginning of the study to allow data collection for up to 14 days. The records were integrated into 60-s epochs, and vector magnitude (VM) data were calculated as a square root of the sum of squared recordings from the accelerometer's three axes. Non-wear periods in accelerometer recordings were identified using Choi's algorithm using the PhysicalActivity package in R [29] and excluded to eliminate the possibility of identifying them as bedrest.

### Selection criteria

To be included in the analysis, a recording required ≥3 valid nights with ≥2 weeknights and ≥1 weekend night. A valid night required ≥ 6 hours of an accelerometer wearing between 10:00 pm and 6:59 am on the next day. Data collected after 48 hours of nonwear were discarded. To obtain a study sample with a uniform age distribution, 100 eligible recordings were

randomly selected from each time point of data collection, producing 400 eligible recordings. Each recording came from a different child to minimize non-independence or clustering within the dataset.

## Visual identification of bedrest and wake periods

Although it is a slow and time-consuming process, visual identification is recommended as the best referent heuristic method for identifying bedrest/sleep and wake time using accelerometry data [2,3]. This study used two trained raters' visual identification as the benchmark for comparing the other *bedrest* identification methods. In this study, we used the terms *bedrest* and *wake* to define sustained periods below or above the optimal threshold of body movements (i.e., counts per min). In the current pediatric literature, terms used for similarly defined and assessed sleep-related periods of absent or minimal activity include "bedrest"[25,30], "sleep" [10,16], "sleep-period time", "nocturnal sleep" [2], and "in-bed time" [3].

Visually identified *bedrest* was defined as prolonged periods (>30min) below or equal to the threshold of 100 counts/min. Each record was viewed using a labeling software developed for this study (available at https://github.com/shi-xin/actigraph_labeler) with two independent raters being able to see each day's accelerometer recordings as VM counts on the vertical axis and time of day in 1-min increments on the horizontal axis. The software enabled the rates to discard data at the beginning and end of each Actigraph file. Raters were trained on heuristics, guiding the visual identification process to estimate each bedrest period's start and end. The guidelines illustrated in Fig 1 were similar to those used in other studies [3,16] and our previous experience.

## Development of DT for bedrest and wake identification

The DT algorithm was developed to automate the process of identifying *bedrest* and *wake* periods from accelerometer recordings. It finds periods with minimal activity (i.e., below a threshold) and then adjusts the periods' beginning and endpoints. This study adapted an automated DT-Youth previously developed for 10-18-yr-old children [25] using a Nelder-Mead method [31] to optimize four DT parameters, named *block length*, *threshold*, *bedrest-end trigger*, *and bedrest-start trigger*. The *block length* is defined as the number of epochs over which an average number of counts per epoch is calculated (e.g., an epoch is 1 min, and a block has 60 epochs, then *block length* is 60 min). *Threshold* (VM counts/min) is the number of counts/block when searching for transition time points from *bedrest* to *wake* or from *wake* to *bedrest*. *Bedrest-start trigger* is the maximum number of VM courts/min needed for a transition from *wake* to *bedrest* or *bedrest* to start. *Bedrest-end trigger* is a minimum number of VM counts/min needed for *wake* to start.

## DT description

The DT has four steps, illustrated in Fig 2. In Step 1, DT divides the data from the accelerometer recordings into blocks (e.g., 60 min), calculates the average counts per epoch in each block. Next, DT determines if the counts/min average in the first block (e.g., 60 min) is lower, equal to, or higher than the tested *threshold* (VM counts/min) to determine its initial status (i.e., *bedrest* or *wake*) of this block and whether to proceed to Step 2 or Step 3. Although categorization is not made by the block, implicitly, blocks with averages below the *threshold* are identified as *bedrest*, while those with averages above are identified as *wake*. Blocks near transitions may be identified as part *bedrest* and part *wake*.

The DT proceeds to Step 2 if the first block is identified as *wake* in Step 1 or after identifying bedrest-start in Step 3. Step 2 identifies bedrest-start. First, DT searches through the next

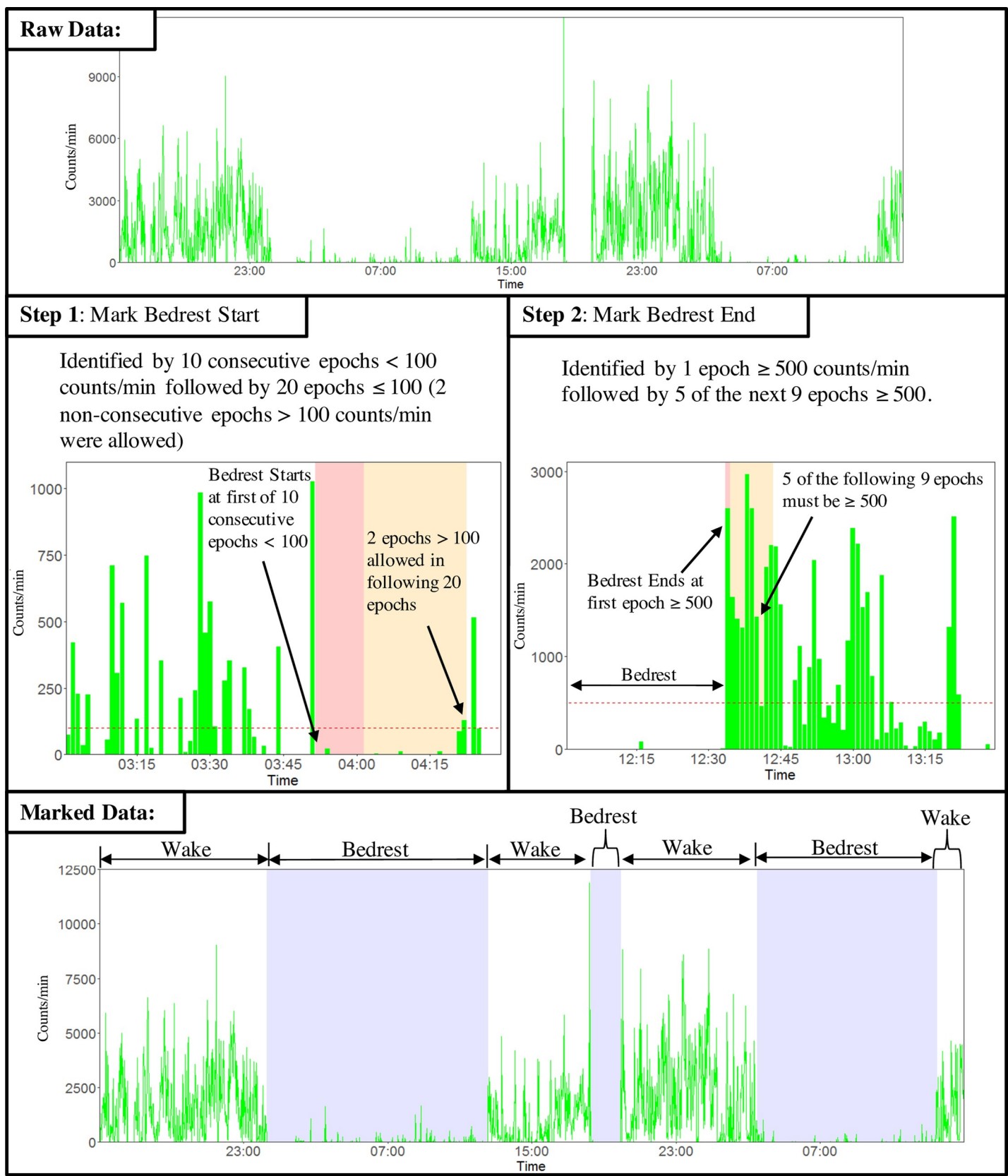

**Fig 1. Graphical presentation of heuristic guidelines for visual identification of accelerometer recordings to estimate each bedrest period's start and end.** The smallest period to be considered a separate episode was 30 min for bedrest and 10 min for awake. Identification guidelines for marking bedrest-start required: (i) 10 consecutive minutes with ≤100 counts/min; (ii) followed by a 20-min period with consistent readings with ≤100 counts/min, but allowing for two minutes >100 counts/min. The first minute with ≤100 counts/min, identified in (i), was marked bedrest-start. Bedrest-end identification required: (i) identifying a minute with ≥500 counts/min; (ii) followed by 5 out of the next 9 minutes with ≥500 counts/min. The minute preceding the first minute with ≥500 counts/min, identified in (i), was marked bedrest-end. The period between bedrest-start and the next bedrest-end was marked *bedrest*, and the period between bedrest-end and the next bedrest-start was marked *wake*. The agreement between raters was assessed using kappa-statistics. To handle a disagreement between the raters in identifying 1-min epochs as bedrest or wake, we generated two separate sets of data named *True1* and *True2*, respectively, for both the development and the validation group. In *True1*, the epochs in which there was disagreement were assumed to be truly *wake*, and in *True2*, they were assumed to be truly *bedrest*. Separate analyses were conducted for both *True1* and *True2*. We report the *True2* analysis as the algorithm was less accurate relative to *True1*. Thus, we report the more conservative estimate of identifying *bedrest*.

block's remaining data with an average value below the *threshold*. Next, DT searches within the identified block and the preceding block for the last two consecutive epochs with VM counts higher than the *bedrest-start trigger* parameter. The epoch after these two epochs is marked bedrest-start.

The DT proceeds to Step 3 if the first block is categorized as *bedrest* in Step 1, or after identifying bedrest-start in Step 2. Step 3 identifies the bedrest-end and follows a similar logic to Step 2. DT searches the remaining blocks for a block with an average above the *threshold*.

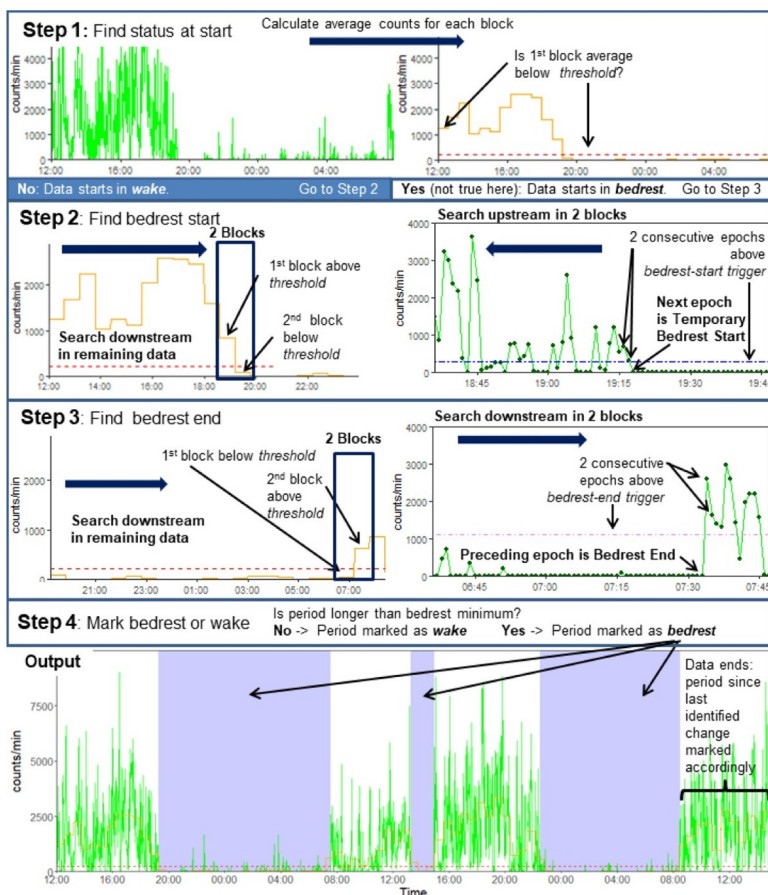

**Fig 2. Simplified decision tree (DT) to identify triaxial accelerometer recordings (counts/epoch) as *bedrest* or *wake*.** The plots contain ~48 h data from a representative child (4-year-old, male) before (Step 1—left panel), during (Step 1 –right panel, Step 2, and Step 3), and after (Step 4) bedrest marking by the DT. The DT uses four algorithm parameter values (*block length*, *threshold*, *bedrest-start trigger, and bedrest-end trigger*) and has a four-step process to run through the data.

Then, DT searches within the identified block and the block proceeding for the epoch that ends bedrest using the *bedrest-end trigger* parameter. The identified blocks are searched for the first two consecutive epochs with VM counts above the trigger. The epoch immediately before these epochs is marked bedrest-end.

Step 4 disregards any *bedrest* period shorter than a user-specified minimum bedrest duration (e.g., 60 min). The DT algorithm cycles between Step 2 identifying bedrest-start and Step 3 identifying bedrest-end until all epochs are marked as either *bedrest* or *wake*. Epochs after the last identified change from *bedrest* to *wake* or *wake* to *bedrest* are marked accordingly.

## Testing DT parameters

An automated Nelder-Mead Simplex Optimization procedure [31] was used to construct and test DT parameter combinations to find combinations that maximized accuracy for *True1* in the development group and another combination that maximized accuracy for *True2* files. The Nelder-Mead algorithm finds the minimum of a real n-dimensional function by generating a simplex with (n+1) vertices. For a two-dimensional function, the simplex would be a triangle, and for three-dimensional, a tetrahedron. The algorithm then adapts the simplex to the local landscape of the function, eventually contracting to a minimum. The algorithm requires four scalar parameters for reflection ($\rho$), expansion ($\chi$), contraction ($\gamma$) and shrinkage ($\alpha$) of the simplex and standard values were used for each; 1, 2, 0.5, and 0.5, respectively. The procedure was initialized with four sets of four parameter values (see S1 Table). Initial testing showed that the procedure took around 80 iterations to converge on a minimum point; therefore, the algorithm was allowed to proceed for 100 iterations in all the subsequent runs to allow space for error. A graphic summary of the optimization process is in S1 Fig, and selected performance parameters for each iteration are in S2 Table.

For each combination of parameters and each participant, every epoch (1 min) in the monitoring wear period was classified by DT as *bedrest* or *wake* and compared to the time-synchronized *bedrest* or *wake* classification from the visual identification. To calculate sensitivity, specificity, and accuracy, every epoch (n = 4,730,734) was placed into one of four categories: (i) true positive (TP), DT, and visual identification agreed on bedrest; (ii) a true negative (TN), DT and visual identification agreed on wake; (iii) a false positive (FP), DT identified as bedrest and visual identification classified as wake; and (iv) a false negative (FN), DT identified as wake and visual identification classified as bedrest [30].

Sensitivity was defined as the proportion of bedrest epochs correctly identified as *bedrest* by the DT (TP/(TP+FN)), and specificity was defined as the proportion of wake epochs correctly identified as *wake* by the DT (TN/(TN+FP)). Accuracy was defined as the proportion of all epochs correctly identified as *bedrest* or *wake* by the DT ((TP + TN)/(TP + TN + FN + FP)). Specificity and sensitivity were considered equally important. For each combination of *block lengths*, *threshold*, *bedrest-start trigger*, and *bedrest-end trigger*, means of sensitivity, specificity, and accuracy were calculated for *True1* and *True2* datasets. To obtain conservative validity estimates, the optimal algorithm parameters for the least accurate approach, *True2*, were recorded and tested on the validation group.

## Assessing DT performance

The DT performance in the validation group was assessed by comparing its accuracy to the mentioned earlier DT-Youth [25] and the automated Sadeh's sleep detection algorithm [26] in ActiLife software (ActiGraph, Pensacola, FL). We used DT-Youth to compare it head-to-head to determine the impact of new parameters on preschool children's DT performance.

We used Sadeh's algorithm because no validated algorithm is available to identify bedrest or sleep for preschool children. Sadeh's algorithm was initially developed in 10-16-yr-old children and 20-25-yr-old adults for wrist-worn accelerometers, but it has been validated for waist-worn accelerometers in 10-18-yr-old children [2,13]. DT and Sadeh's algorithm's significant difference is that DT allows a user-specified minimum period to be identified as bedrest (e.g., 30 min). Sadeh's algorithm classifies each 1-min epoch, either sleep or wake and summarizes sleep epochs to assess the total period of sleep. As recommended by others [2,32,33], DT classification (*bedrest and wake*) was synchronized minute-by-minute and compared with visual identification (*bedrest* or *wake*), DT-Youth (*bedrest* or *wake*), and Sadeh's classification (*sleep* or *wake*).

## Bedrest and wake periods classification

In the validation group, bedrest periods lasting <30 min were classified as wake, periods lasting from ≥30 to ≤90 min were classified as naps, and periods lasting >90 min were classified as bedrest. Naps frequency and duration were calculated as an average number and total duration of naps per day and included in the total bedrest analyses. The 30-min lower threshold for bedrest was selected by assuming that children are typically in bed for more than 30 min for their daytime naps. The 90-min upper threshold for naps was selected by assuming that bedrest longer than 90 min and following or preceding a wake period should be classified as a distinct bedrest episode. Awakenings during bedrest were defined as wake periods lasting from ≥5 to ≤15 min (e.g., to visit the bathroom) during bedrest lasting >90 min and are presented as an average number and total duration of awakenings per day.

## Parent-reported sleep survey

The parent-reported sleep survey conducted at the start of the accelerometer wearing period was used to obtain the usual child's bedtime and wake time and the frequency and duration of daytime naps. Survey sleep time was calculated as the sum of reported nap duration and the elapsed time between the reported bedtime and wake time. It was compared with DT and Sadeh's algorithm assessment of bedrest or sleep duration, respectively.

## Statistical analysis

From the group of selected participant recordings (n = 400), 200 were randomly allocated to the development and 200 to validation groups using the random sampling function in R. The development group was used to optimize DT algorithm parameters values (*block length*, *threshold, bedrest-start trigger, and bedrest-end trigger*). The differences between the groups in age and anthropometric measures were compared using the independent sample t-test and chi-square, respectively. The agreement between raters in the visual identification of bedrest was calculated using a weighted Cohen's Kappa coefficient [34].

The sensitivity, specificity, and accuracy of the DT algorithm were compared for agreement with the visual identification. The differences in sensitivity, specificity, and accuracy between the development and validation groups were compared using independent samples t-test. In the validation group, differences between DT and Sadeh's algorithm, between DT and parental survey, between Sadeh's algorithm and parental survey, between DT bedrest identification. For these calculations, we used the VM counts data. Since Sadeh's algorithm uses a single (vertical) axis data from an accelerometer, we also compared the difference between Sadeh's algorithm and DT using recordings from the vertical axis using paired samples t-test. Bedrest characteristics were performed after epochs (minutes) of not-wearing and discarded in visual identification were removed. The agreement between the parental survey and other methods

in the validation group was assessed using overlap coefficients calculated for bedrest identified by DT and parental survey, Sadeh's and parental survey, and DT and Sadeh's algorithm. Overlap was defined as the time both methods agreed on bedrest/sleep identification divided by the shortest time of total sleep/bedrest identified by either method. Therefore, if the sleep period identified by one method was a subset of that identified by the other method, the overlap coefficient would be equal to 1.0. Overlap coefficients were calculated for each child in the validation group, and the average is reported.

All results are presented as means, standard deviations (SD), ranges, and statistical significance set at P < 0.05. The programming language R version 3.6.1 was used to develop DT and perform statistical analyses.

## Results

### Participants' characteristics

The study population (n = 400, 51.5% male) included Hispanic/Latino (n = 361, 90.3%), non-Hispanic black (n = 24, 6.0%), non-Hispanic white (n = 9, 2.3%), and non-Hispanic multi-racial (n = 6, 1.5%) children. Of the eligible children, 250 (62.5%) were normal weight (<85[th] BMI percentile), 111 (27.8%) were overweight (≥85[th] to <95[th] BMI percentile), and 39 (9.8%) were obese (≥95[th] BMI percentile). Among parents, 246 (61.5%) had less than a high school degree or equivalent. The average child age was 5.03 ± 1.14 years, weight was 19.68 ± 3.69 kg, height was 107.90 ± 8.64 cm, and BMI was 16.78 ± 1.12 kg/m$^2$ (see Table 1). There were no significant differences in personal characteristics between participants in the development and validation groups (all P>0.05).

### Agreement between raters in visual bedrest identification

The inter-rater agreement was very good (mean sensitivity, specificity, and accuracy were > 0.98 and $\kappa$ (kappa) = 0.99).

### DT algorithm optimal parameters

Within the development group, the optimal *block length* was 36 min, the *threshold* was 230 counts/min, the *bedrest-start trigger* was 305 counts/min, and the *bedrest-end trigger* was 1,129 counts/min. These parameter values are default in the PhysActBedRest package, but the end-

**Table 1. Characteristics of study participants.**

| | All Participants (n = 400) | Development Group (n = 200) | Validation Group (n = 200) | P-value |
|---|---|---|---|---|
| Male \| Female, n (%) | 206 \| 194 (51.50 \| 48.50) | 108 \| 92 (54.00 \| 46.00) | 98 \| 102 (49.00 \| 51.00) | 0.317[a] |
| Age (years) | 5.03 ± 1.14 (3.01, 6.98) | 5.00 ± 1.11 (3.01, 6.98) | 5.05 ± 1.18 (3.01, 6.94) | 0.707[b] |
| Height (cm) | 107.90 ± 8.64 (88.80, 129.40) | 107.82 ± 8.63 (90.45, 128.65) | 107.90 ± 8.66 (88.80, 129.40) | 0.933[b] |
| Weight (kg) | 19.68 ± 3.69 (13.00, 40.40) | 19.61 ± 3.66 (13.30, 31.80) | 19.74 ± 3.74 (13.00, 40.40) | 0.718[b] |
| Body Mass Index (kg/m$^2$)[c] | 16.78 ± 1.12 (14.19, 24.11) | 16.73 ± 1.07 (14.19, 21.02) | 16.83 ± 1.16 (14.44, 24.11) | 0.387[b] |
| Accelerometer wear (days) | 8.32 ± 1.74 (4.00, 20.00) | 8.31 ± 1.89 (4.00, 20.00) | 8.34 ± 1.60 (5.00, 15.00) | 0.841[b] |
| Accelerometer wear (hours/day) | 20.86 ± 2.37 (10.92, 23.99) | 21.09 ± 2.23 (12.67, 23.97) | 20.72 ± 2.51 (10.92, 23.99) | 0.250[b] |

Values are mean ± standard deviation and values in parentheses are ranges.

[a] Pearson's chi-square test between development and validation groups.

[b] independent samples t-test between development and validation groups.

[c] body mass index (BMI).

**Table 2. DT performance in the development and validation groups.**

|  | Development group (n = 200) | Validation group (n = 200) | P-value[a] |
|---|---|---|---|
| Sensitivity | 0.948 ± 0.058 (0.501, 0.998) | 0.941 ± 0.056 (0.618, 0.999) | 0.258 |
| Specificity | 0.976 ± 0.028 (0.798, 0.998) | 0.974 ± 0.038 (0.566, 0.999) | 0.414 |
| Accuracy | 0.961 ± 0.042 (0.626, 0.995) | 0.956 ± 0.043 (0.654, 0.994) | 0.305 |

Values are mean ± standard deviation and values in parentheses are ranges.

[a] independent samples t-test between development and validation groups.

user can easily change them if they wish (e.g., for use in preschool children with sleep disturbances).

## DT performance

Using visual identification as the benchmark, differences between the development and validation groups were not statistically significant for sensitivity, specificity, and accuracy (all P>0.05, see Table 2).

In the validation group, DT performed better at identifying *bedrest* than DT-Youth and Sadeh's algorithm measured by sensitivity, specificity, and accuracy for VM counts data (P<0.001; see Table 3).

The DT performed better when using VM than single-axis (vertical) recordings from accelerometers (P<0.001, see S3 Table). Also, DT performed better than Sadeh's algorithm when both were using data from single-axis (vertical) data (P<0.001, see S4 Table).

## Bedrest characteristics in the validation group

As identified by DT and shown in Table 4, the average daily bedrest time was over 9 hours (502 ± 62 min/day), ranging from almost 5 hours (294 min/day) to over 10 hours (652 min/day). Mean sleep duration obtained from the parental survey was significantly longer than bedrest identified by DT (650 and 502 min/day; P<0.001), Sadeh's algorithm (650 and 604 min/day; P<0.001), and visual identification (650 and 502 min/day; P<0.001).

On average, the wake time identified by DT was over 11 hours (734 min/day) and was longer than Sadeh's algorithm (693 min/day),. The number of children taking naps reported in the parental survey (n = 45) was lower than the DT identified (n = 134). Naps were identified by DT in 67% of the children at least once during the monitoring period. In children whose parents reported taking daytime naps daily (23%), the naps frequency was higher than those identified by DT (1.05 ± 0.27 and 0.25 ± 0.24). The parental survey's average naps length duration was approximately 25 min longer than that assessed by DT. The average number of

**Table 3. Performance of DT, DT-Youth, and Sadeh's algorithm in the validation group.**

|  | DT (n = 200) | DT -Youth (n = 200) | Sadeh's (n = 200) | P-value [a,b] |
|---|---|---|---|---|
| Sensitivity | 0.941 ± 0.056 (0.618, 0.999) | 0.855 ± 0.086 (0.558, 0.999) | 0.920 ± 0.041 (0.706, 0.979) | < 0.001 |
| Specificity | 0.974 ± 0.038 (0.566, 0.999) | 0.870 ± 0.079 (0.564, 0.999) | 0.888 ± 0.064 (0.506, 0.984) | < 0.001 |
| Accuracy | 0.956 ± 0.043 (0.654, 0.994) | 0.861 ± 0.062 (0.588, 0.979) | 0.902 ± 0.042 (0.599, 0.965) | < 0.001 |

Values are mean ± standard deviation and values in parentheses are ranges.

[a] paired t-test between DT algorithm and Sadeh's algorithm.

[b] paired t-test between DT algorithm and DT-Youth algorithm.

**Table 4. Characteristics of bedrest and wake periods identified using DT, Sadeh's algorithm, and parental survey, and visual identification in the validation group (n = 200).**

| | DT | Sadeh's algorithm | Parental survey | Visual identification | P-value |
|---|---|---|---|---|---|
| Bedrest [min/day] | 502 ± 62 (294, 652) | 556 ± 73 (301, 847) | 650 ± 81 (450, 960) | 502 ± 63 (302, 664) | <0.001[a,b,d,e,f] 0.859 [c] |
| Wake [min/day] | 734 ± 101 (291, 898) | 693 ± 88 (286, 850) | - | 733 ± 103 (292, 937) | <0.001[a,e] 0.859 [c] |
| Children taking naps [n (%)] | 134 (67.0) | - | 45 (22.5) | - | <0.001[b] |
| Nap frequency [g] [number/day] | 0.25 ± 0.24 (0.00 [h], 1.00) | - | 1.05 ± 0.27 (1.00, 3.00) | - | <0.001[b] |
| Nap duration [g] [min/day] | 47 ± 34 (0 [h], 84) | - | 72 ± 38 (15, 180) | - | <0.001[b] |
| Awakenings during bedrest [i] [number/day] | 1.8 ± 0.8 (0.4, 4.4) | - | - | - | |
| Awakenings during bedrest [min/day] | 8.3 ± 1.9 (5, 15) | - | - | - | |

Values are mean ± standard deviation and values in parentheses are ranges.

[a] paired t-test between DT and Sadeh's algorithm.

[b] paired t-test between DT and Parental survey.

[c] paired t-test between DT and Visual identification.

[d] paired t-test between Sadeh's algorithm and Parental survey.

[e] paired t-test between Sadeh's algorithm and Visual identification.

[f] paired t-test between Parental survey and Visual identification.

[g] nap: 30 min ≤ bedrest ≤ 90 min in children whose parents reported taking a nap (n = 45).

[h] naps were not identified by DT in all children whose parent reported taking naps (n = 14, min/day = 0).

[i] awakening: 5 min ≤ wake ≤ 15 min.

bedrest awakenings identified by DT was 1.8 per day, ranging from 0.4 to 4.4. The bedrest awakenings duration was 8.3 min/day and ranged from 5 to 15 min/day.

The overlap coefficients for sleep duration between parental survey and DT and parental survey and Sadeh's algorithm were 0.860 and 0.843, respectively (see Table 5).

## Discussion

In this study, a DT algorithm previously developed to identify bedrest periods using waist-worn accelerometers recordings for youth (10-18-yr-old) [25] was adapted for use in preschool children. The adapted DT provided very good sensitivity, specificity, and accuracy (all >0.95) for identifying *bedrest* and *wake* when using a visual identification method as the benchmark.

This study did not attempt to assess physiological sleep, but rather the goal was to identify periods of inactivity as *bedrest*, which most likely included nighttime sleep lasting > 90 min and daytime naps lasting from 30 to 90 min. In the validation group, we compared the modified DT performance for preschool children with the DT with parameters established for youth [25]. In a head-to-head comparison, we found that the modified DT performed better in assessing bedrest than DT-Youth. The optimal algorithm parameters for preschool children

**Table 5. Overlap in sleep periods assessed from the parental survey, DT (decision tree) identification, and Sadeh's algorithm in the validation group (n = 191).**

| | Overlap Coefficient | P-value[a] |
|---|---|---|
| Parental vs. DT | 0.860 ± 0.071 (0.524, 0.993) | **< 0.001** |
| Parental vs. Sadeh's | 0.843 ± 0.063 (0.547, 0.968) | **< 0.001** |
| DT vs. Sadeh's | 0.930 ± 0.018 (0.876, 0.973) | **< 0.001** |

Values are mean ± standard deviation and values in parentheses are ranges.

[a] paired t-test between total sleep of sleep assessment method.

(3-6-year-old) differed from values for youth (10-18-yr-old), underscoring a need for age-specific algorithm parameters to identify bedrest and wake accurately.

In the past, several studies utilized automatic algorithms to separate sleep and wake based on recordings from a single accelerometer axis [2,26,33]. The current study took advantage of technological advancements and used VM calculated from counts recorded by three distinct axes to measure movement. We compared the DT performance to identify bedrest for VM and single-axis (vertical) recordings in the validation group and found that both sensitivity and specificity were higher for VM than single-axis data. We recommend that studies assessing bedrest/sleep in preschool children using triaxial accelerometers utilize VM rather than single-axis data. Other advantages of using the modified algorithm its ability to identify short naps (30–60 min) usually occurring during the day and using data from several days of accelerometer recordings.

Using visual identification as the benchmark, a comparison of the DT to Sadeh's algorithm [26] demonstrated that the DT had higher sensitivity, specificity, and accuracy than Sadeh's algorithm both for VM and single-axis data. DT's better performance than Sadeh's algorithm could be expected since Sadeh's algorithm was developed in a different population and using a different study protocol [26]. A more objective comparison of algorithms' generalization capacity would be achieved using an independent preschool-age children dataset. Another plausible explanation could be that Sadeh's algorithm classified short daytime periods of low activity (e.g., sedentary behavior) as "sleep." The DT protected against this problem by limiting *bedrest* to inactivity lasting more than 30 minutes. However, while imposing this restriction could cause misclassification of short daytime naps as *wake*, the risk is likely low in this sample, as only 2 out of 45 parents reported naps <30 min. The R function developed in this study and readily available to researchers allows the user to set the minimum bedrest period to balance these concerns in their study populations.

We did not compare the DT to other currently available automated algorithms because they were developed and validated in older children [2], used a different methodology [16], or different accelerometers [8,9]. Among automated algorithms validated for waist-worn Actigraph accelerometers, the first was an algorithm developed by Tudor-Locke et al. [35] for 10-18-yr-old children combining visual inspection to mark the onset and offset of nocturnal sleep from accelerometer recorded with the sleep diary data. The newer version of this algorithm [2] showed a moderately high correlation of nocturnal sleep (r = 0.61 to 0.74) with visual assessment of accelerometry data and sleep diaries.

A parental survey was used to calculate the child's average length of sleep/bedrest and nap frequency and duration. Similar to other reports [17,23], this study found that sleep duration assessed from the parental survey was significantly higher than bedrest and sleep assessed by DT and Sadeh's algorithm, respectively. On average, nighttime sleep reported by parents was about an hour longer than bedrest identified by the DT, which can reflect the difference between the time a child was put in bed and the time the child fell asleep. Notably, the bedrest periods assessed by DT overlapped, approximately 86%, with the sleep periods derived from the parental survey.

The frequency and duration of naps obtained from the survey were also different from those identified by DT, with parents reporting longer nap time by an average of 12 minutes. This finding emphasizes the importance of a quantitative approach to capturing bedrest/sleep for young children, rather than relying on the parental report as the sole measure of sleep.

The study had several strengths. First, we used a relatively large (n = 400) and diverse group regarding sex, age, and body mass indices [27]. Random selection of the development and validation groups allowed for the algorithm's robust performance and the avoidance of overfitting due to the small sample size. Second, the study was conducted in field conditions that increased the DT's generalizability compared to laboratory studies. The average time spent in

bedrest ranged from over 5 to almost 12 hours/day, which is similar to the sleep time range reported for a national sample of preschool children in the US [6]. Third, the DT was optimized using objective accelerometer data rather than subjective sleep diary data. In the absence of a practical reference standard criterion method that can be used with 3-6-yr-old children in daily life, we consider the DT a potentially useful approach for separating nighttime sleep and daytime naps from sedentary behaviors and physical activity. Future studies should be performed in diverse populations to test and calibrate the DT against PSG and compare it with other automatic algorithms that use different accelerometry data measures such as arm angle or gravitational acceleration data expressed in grams [16]. Future research could also utilize the DT or similar methodology when examining preschool-age children's sleep patterns and associations with learning and health outcomes and help inform clinical decision making [36].

The study had some limitations. First, this study used waist-worn monitors, which are preferred for measuring physical activity and sedentary behaviors, rather than non-dominant wrist-worn monitors frequently used to assess sleep [19]. However, the evidence shows that waist-worn accelerometers can accurately assess sleep duration in school-age children [13,18,35]. Second, the study results are specific to 3–6 years old, healthy children. However, we believe the set of parameters (block length, threshold, bedrest-start trigger, and bedrest-end trigger) optimized in this study ensures very good DT performance in all preschool children. The DT should also be validated in other studies with an independent sample of preschool-age children and performed in a different environment. However, the DT allows users to adapt it for bedrest identification in various populations and/or using different accelerometers. Third, the DT algorithm validity was not tested in children with sleep disturbances and other clinical pediatric populations and might require additional studies. Forth, accelerometer recordings were collected at 40 Hz and not commonly used 30 Hz. A recent study found an error between 10–100 Hz frequency decreased with declining movement intensity, suggesting that the difference between 40 Hz and 30 Hz would have a small effect on bedrest identification in the current study [37]. Nevertheless, future users of the algorithm should be aware of this limitation.

## Conclusions

The DT-based algorithm originally developed for youth (10-18-year-old) children was adapted for preschool children to identify time spent in nighttime *bedrest* and daytime *naps* using data from waist-worn accelerometers with very good sensitivity, specificity, and accuracy. The automated bedrest/sleep detection DT algorithm is openly accessible as a package for the R software environment ("PhysActBedRest").

## Supporting information

**S1 Fig. Nelder-Mead simplex optimization progression for *True 2* dataset.**
(TIF)

**S1 Table. Parameters sets used to initialize DT (decision tree) algorithm optimization using the Nelder-Mead simplex procedure.**
(DOCX)

**S2 Table. Progression of the optimization of decision tree (DT) algorithm parameters.**
(DOCX)

**S3 Table. Comparison of decision tree (DT) algorithm performance between vector magnitude (VM) and single-axis (vertical) recordings in the validation group (n = 200).**
(DOCX)

**S4 Table. Comparison between performance between decision tree (DT) and Sadeh's algorithm using recordings from the vertical axis in the validation group (n = 200).**
(DOCX)

## Acknowledgments

We like to thank Shi Xin for his programming work and participating in the visual data identification and Dr. Nina Martin and Eli Po'e for help with manuscript review and editing. We also acknowledge the staff and community members' work at the Coleman and Southeast Community Centers in Nashville, Tennessee. With gratitude, we thank them for their commitment to working with our team.

## Author Contributions

**Conceptualization:** J. Dustin Tracy, Shari L. Barkin, Maciej S. Buchowski.

**Data curation:** Shari L. Barkin, Maciej S. Buchowski.

**Formal analysis:** Thomas Donnelly, Evan C. Sommer.

**Funding acquisition:** Shari L. Barkin, Maciej S. Buchowski.

**Investigation:** Evan C. Sommer, Shari L. Barkin, Maciej S. Buchowski.

**Methodology:** J. Dustin Tracy, Evan C. Sommer, Shari L. Barkin, Maciej S. Buchowski.

**Project administration:** Shari L. Barkin.

**Resources:** Shari L. Barkin, Maciej S. Buchowski.

**Software:** J. Dustin Tracy, Thomas Donnelly.

**Supervision:** Shari L. Barkin, Maciej S. Buchowski.

**Validation:** Maciej S. Buchowski.

**Writing – original draft:** J. Dustin Tracy, Evan C. Sommer, Maciej S. Buchowski.

**Writing – review & editing:** J. Dustin Tracy, Thomas Donnelly, Evan C. Sommer, William J. Heerman, Shari L. Barkin, Maciej S. Buchowski.

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
