## [Decision Letter · Decision Letter 0]

2 Nov 2020

PONE-D-20-27484

Identifying bedrest using waist-worn triaxial accelerometers in preschool children

PLOS ONE

Dear Dr. Buchowski,

Thank you for submitting your manuscript to PLOS ONE. After careful consideration, we feel that it has merit but does not fully meet PLOS ONE’s publication criteria as it currently stands. Therefore, we invite you to submit a revised version of the manuscript that addresses the points raised during the review process.

Please address all comments by the reviewer.

We look forward to receiving your revised manuscript.

Kind regards,

Raffaele Ferri, MD

Academic Editor

PLOS ONE

Journal Requirements:

2. Please ensure that you refer to Figure 3 in your text as, if accepted, production will need this reference to link the reader to the figure.

"This research was supported by the National Institutes of Health (NIH) grants HL103620 and HL103561 from the National Heart, Lung, and Blood Institute and the Eunice Kennedy Shriver National Institute of Child Health and Development and the Office of Behavioral and Social Sciences Research: https://www.nih.gov/. Additional funds were provided by grants DK058404 and DK20593 from the National Institute of Diabetes and Digestive and Kidney Diseases: https://www.nih.gov. The REDCap Database was supported by TR000445 from the National Center for Advancing Translational Sciences: https://www.nih.gov/. WJH was supported by K23 HL127104 grant from the National Heart Lung and Blood Institute: https://www.nih.gov/.

i) Please provide an amended statement that declares *all* the funding or sources of support (whether external or internal to your organization) received during this study, as detailed online in our guide for authors at http://journals.plos.org/plosone/s/submit-now.  Please also include the statement “There was no additional external funding received for this study.” in your updated Funding Statement.

ii) Please include your amended Funding Statement within your cover letter. We will change the online submission form on your behalf.

Reviewers' comments:

Reviewer's Responses to Questions

**Comments to the Author**

1. Is the manuscript technically sound, and do the data support the conclusions?

Reviewer #1: Partly

2. Has the statistical analysis been performed appropriately and rigorously? 

Reviewer #1: Yes

3. Have the authors made all data underlying the findings in their manuscript fully available?

Reviewer #1: Yes

4. Is the manuscript presented in an intelligible fashion and written in standard English?

Reviewer #1: Yes

5. Review Comments to the Author

Reviewer #1: Authors have shared an interesting article about the validation of an adapted algorithm for identification and differentiation of bedrest and wake periods using wait-worm triaxial accelerometers. The work is well structured and includes comprehensive description of methodology and statistical analyses.

My biggest concern regards to the fairness of the comparison approach with other methods, in particular to what respects to the omission of the role that the database variability effect might play on the results, possibly penalizing performance of the other methods. In effect, authors openly admit, and experimental results also confirm, that usage of the proposed DT(-Youth) algorithm on different age groups (in this case, preschool vs older children) does require reconfiguration, namely different values for the four parameters (block length, threshold, bedrest-start trigger, and bedrest-end trigger) guiding the decision algorithm. Given for example, the benchmarked Sadeh´s algorithm was probably developed based on data other than the Growing Right Onto Wellness dataset used in this study (meaning different subjects and signal characteristics), it is somehow expected that performance could drop when comparing it with an algorithm that was optimized for the mentioned dataset.

A more fair comparison would be to use a third, completely independent dataset to all the involved algorithms in the comparison, to test the real generalization capabilities of each one.

Authors should develop into this point, and at the very least include the issue into the discussion and address it as a possible limitation.

Authors should also elaborate more on the specific differences between the current version of the DT algorithm proposed and the previously reported, here referred as DT-Youth. It is just about finding a different set of configuration parameters optimized for each targeted age group for the same algorithm? Or are the two methods intrinsically different?

Necessity to reparametrize the algorithm on each case could be seen as an additional limiting factor. Apparently every potential user would have to run an optimization procedure similar to the one described by the authors to discover the optimal parameters for their specific local database, which seems rather unpractical, unless any general rule-of-thumb can be derived a priori to find an appropriate set.

An additional point that requires further clarification is that one of the main reasons motivating this work is that, to the author´s knowledge, “no accelerometry-based algorithm validation studies in healthy preschool children have been reported” (lines 77-78). What are the reasons for which authors have opted not to include the work referenced in [9] into this discussion?

Minor comments:

Though Table 4 suggest that no abrupt differences exist in the balance of cases among the two categories considered (bedrest, wake), it would be nice to mention the exact amount of cases on the development and validation subsets, according to the gold standard, in this case, the expert’s classifications. If classes would result unbalanced then adding kappa to the reported list of performance statistics might be a better choice than accuracy, as kappa compensates for agreement due to chance, i.e. in case the a priori probability of one class if significantly bigger than the other. It would also allow comparison of the kappa agreement reported to evaluate the expert’s inter-scorer reliability.

There is possibly a typo when referencing to Fig. 2 in line 334, which I think it should be Fig. 3 instead.

In addition, I would suggest reconsidering Fig.3 in general and stick reporting to the Area Under to ROC Curve for the final optimum point for the different algorithms, as there is no much interest in reporting the intermediate values during the optimization process. Usually one reports ROC curves to compare different operating final points of an algorithm. Regardless, if decided to report (or display) the results in both the development and validation subsets, identify them more clearly. Now assuming the big plot corresponds to development and the smaller one to the validation set.

6. PLOS authors have the option to publish the peer review history of their article (what does this mean?). If published, this will include your full peer review and any attached files.

Reviewer #1: No

---

## [Author Response · Author response to Decision Letter 0]

23 Dec 2020

Reviewer #1: Authors have shared an interesting article about the validation of an adapted algorithm for identification and differentiation of bedrest and wake periods using wait-worm triaxial accelerometers. The work is well structured and includes comprehensive description of methodology and statistical analyses.

Response: Thank you very much.

1. My biggest concern regards to the fairness of the comparison approach with other methods, in particular to what respects to the omission of the role that the database variability effect might play on the results, possibly penalizing performance of the other methods. In effect, authors openly admit, and experimental results also confirm, that usage of the proposed DT(-Youth) algorithm on different age groups (in this case, preschool vs older children) does require reconfiguration, namely different values for the four parameters (block length, threshold, bedrest-start trigger, and bedrest-end trigger) guiding the decision algorithm. Given for example, the benchmarked Sadeh's algorithm was probably developed based on data other than the Growing Right Onto Wellness dataset used in this study (meaning different subjects and signal characteristics), it is somehow expected that performance could drop when comparing it with an algorithm that was optimized for the mentioned dataset. A more fair comparison would be to use a third, completely independent dataset to all the involved algorithms in the comparison, to test the real generalization capabilities of each one.

Authors should develop into this point, and at the very least include the issue into the discussion and address it as a possible limitation.

Response. We agree with the Reviewer. Having an independent dataset would be an excellent addition to the study, but we did not have such a dataset available. As the Reviewer suggested, we added this concern to the study limitations. We also included the Reviewer's suggestion as one of the directions for future studies. 

Lines 422-425. Better performance of DT than Sadeh's algorithm could be expected since Sadeh's algorithm was developed in a different population and using a different study protocol [26]. A more objective comparison of algorithms' generalization capacity would be achieved using an independent preschool-age child dataset.

Lines 476-479. The DT should also be validated in other studies with an independent sample of preschool-age children performed in a different environment. However, the DT allows users to adapt it for bedrest identification in various populations and/or using different accelerometers.

Lines 464-467. Future studies should be performed in diverse populations to test and calibrate the DT against PSG and compare it with other automatic algorithms that use different accelerometry data measures such as arm angle or gravitational acceleration data expressed in grams [16].

2. Authors should also elaborate more on the specific differences between the current version of the DT algorithm proposed and the previously reported, here referred as DT-Youth. It is just about finding a different set of configuration parameters optimized for each targeted age group for the same algorithm? Or are the two methods intrinsically different?

Response. As the Reviewer pointed out, the DT algorithm is based on the DT-Youth. The significant enhancements are using vector magnitude (VM) and using data from several days of recordings.

Lines 411-419. The current study took advantage of technological advancements and used VM calculated from counts recorded by three distinct axes to measure movement. We compared the DT performance to identify bedrest for VM and single-axis (vertical) recordings in the validation group and found that both sensitivity and specificity were higher for VM than single-axis data. We recommend that studies assessing bedrest/sleep in preschool children using triaxial accelerometers utilize VM rather than single-axis data. Another advantage of using the modified algorithm is its ability to identify short naps (30-60 min) usually occurring during the day and using data from several days of accelerometer recordings.

3. Necessity to reparametrize the algorithm on each case could be seen as an additional limiting factor. Apparently every potential user would have to run an optimization procedure similar to the one described by the authors to discover the optimal parameters for their specific local database, which seems rather unpractical, unless any general rule-of-thumb can be derived a priori to find an appropriate set.

Response. The study's goal was to modify the existing algorithm that could be used in healthy 3-6-year-old children. We believe that the algorithm we propose offers robust performance in healthy preschool children. Using the set of parameters (block length, threshold, bedrest-start trigger, and bedrest-end trigger) has been validated in youth (ref #25) and adults (ref #30). The optimal values for these parameters are default in the algorithm's package. However, the program allows investigators to change the parameters' value if they choose. 

Lines 473-479: Second, the study results are specific to 3-6 years old, healthy children. However, we believe the set of parameters (block length, threshold, bedrest-start trigger, and bedrest-end trigger) optimized in this study ensures very good DT performance in all preschool children. The DT should also be validated in other studies with an independent sample of preschool-age children and performed in a different environment. However, the DT allows users to adapt it for bedrest identification in various populations and/or using different accelerometers.

4. An additional point that requires further clarification is that one of the main reasons motivating this work is that, to the authors' knowledge, no accelerometry-based algorithm validation studies in healthy preschool children have been reported" (lines 77-78). What are the reasons for which authors have opted not to include the work referenced in [9] into this discussion?

Response. Thank you for reading our manuscript in detail. The reason for not including Belanger et al. (reference 9) study was that it was performed using an Actiwatch accelerometer that uses recordings from a single axis. The results (counts) are not comparable with results from other accelerometers, including tri-axial accelerometers. In other words, Belanger’s algorithm(s) cannot be used directly for processing data from the Actigraph-GT3X accelerometer used in our study. We used Actigraph because it generates data in g-units, and it is triaxial, allowing the calculation of vector magnitude (VM) useful in assuming physical activity. Actigraph is used in many US studies in children and adults, including the National Health and Nutrition and Examination Survey (NHANES). Relatively widespread use of this accelerometer makes it easier to compare results among studies. 

We clarified the statement and included additional information about reference 9 (Belenger et al. 2013).

Lines 79-85: Recently, Belanger et al. [9] developed two algorithms for the wrist-worn and ankle-worn single-axis Actiwatch accelerometer using data from 2-5 years old children (n=12) during nighttime sleep. However, no triaxial accelerometer-based algorithm validation studies in healthy preschool children have been reported to our knowledge. The present study aims to fill a gap in the literature by identifying bedrest periods that include nighttime sleep and daytime naps from data from accelerometers collected for several days with a high degree of accuracy.

Minor comments:

1. Though Table 4 suggest that no abrupt differences exist in the balance of cases among the two categories considered (bedrest, wake), it would be nice to mention the exact amount of cases on the development and validation subsets, according to the gold standard, in this case, the expert's classifications. If classes would result unbalanced then adding kappa to the reported list of performance statistics might be a better choice than accuracy, as kappa compensates for agreement due to chance, i.e. in case the a priori probability of one class if significantly bigger than the other. It would also allow comparison of the kappa agreement reported to evaluate the expert's inter-scorer reliability.

Response. As suggested, we added information about bedrest and wake duration from the visual identification to Table 4 (the relevant part of the table is included below). The differences between the visual classification, DT, Sadeh’s, and Parental survey did not suggest any unexpected differences between the methods. Therefore, we did not perform further statistical analyses. However, if the Reviewer would recommend performing additional statistical analyses, we could include such analyses either in the manuscript or supplementary materials. Also, the agreement between the raters was assessed using accuracy and kappa coefficient, and the agreement was very good.

Lines 325-326: The inter-rater agreement was very good (mean sensitivity, specificity, and accuracy were >0.98 and κ (kappa) = 0.99).

Table 4. Characteristics of bedrest and wake periods identified using DT, Sadeh’s algorithm, and parental survey in the validation group (n=200).

 DT 

 Sadeh’s algorithm Parental survey Visual identification P-value

Bedrest

[min/day] 

564 ± 77

(379, 719) 

604 ± 80

(379, 813) 

650 ± 81

(450, 960) 

575 ± 80

(287, 734) 

<0.001a,b,c,d,e,f

Wake

[min/day] 680 ± 112

(24, 876) 640 ± 112

(181, 154) - 668 ± 124

(199, 992) <0.001a,c,e

2. There is possibly a typo when referencing to Fig. 2 in line 334, which I think it should be Fig. 3 instead.

Response. Thank you. We did not correct this error because Figure 3 was omitted from the revised manuscript, as suggested by the Reviewer in minor comment 3 (below).

3. In addition, I would suggest reconsidering Fig.3 in general and stick reporting to the Area Under to ROC Curve for the final optimum point for the different algorithms, as there is no much interest in reporting the intermediate values during the optimization process. Usually one reports ROC curves to compare different operating final points of an algorithm. Regardless, if decided to report (or display) the results in both the development and validation subsets, identify them more clearly. Now assuming the big plot corresponds to development and the smaller one to the validation set.

Response. We agree. We have omitted Figure 3 in the revised manuscript.

Editorial comments.

Response. The manuscript meets PLOS ONE's style requirements.

2. Please ensure that you refer to Figure 3 in your text as, if accepted, production will need this reference to link the reader to the figure.

Response. Figure 3 was omitted in the revised manuscript, as recommended by the Reviewer.

Response. The relevant data files (with DOIs) are now available in a public repository:

https://figshare.com/articles/dataset/Train_Files/12904124

https://figshare.com/articles/dataset/Test_Files/12904127

"This research was supported by the National Institutes of Health (NIH) grants HL103620 and HL103561 from the National Heart, Lung, and Blood Institute and the Eunice Kennedy Shriver National Institute of Child Health and Development and the Office of Behavioral and Social Sciences Research: https://www.nih.gov/. Additional funds were provided by grants DK058404 and DK20593 from the National Institute of Diabetes and Digestive and Kidney Diseases: https://www.nih.gov/. The REDCap Database was supported by TR000445 from the National Center for Advancing Translational Sciences: https://www.nih.gov. WJH was supported by K23 HL127104 grant from the National Heart Lung and Blood Institute: https://www.nih.go/.

i) Please provide an amended statement that declares *all* the funding or sources of support (whether external or internal to your organization) received during this study, as detailed online in our guide for authors at http://journals.plos.org/plosone/s/submit-now. Please also include the statement “There was no additional external funding received for this study.” in your updated Funding Statement.

ii) Please include your amended Funding Statement within your cover letter. We will change the online submission form on your behalf.

Response. Updated funding statement is included in the cover letter.

---

## [Decision Letter · Decision Letter 1]

6 Jan 2021

PONE-D-20-27484R1

Identifying bedrest using waist-worn triaxial accelerometers in preschool children

PLOS ONE

Dear Dr. Buchowski,

Thank you for submitting your manuscript to PLOS ONE. After careful consideration, we feel that it has merit but does not fully meet PLOS ONE’s publication criteria as it currently stands. Therefore, we invite you to submit a revised version of the manuscript that addresses the remaining minor points raised during the review process.

We look forward to receiving your revised manuscript.

Kind regards,

Raffaele Ferri, MD

Academic Editor

PLOS ONE

Additional Editor Comments (if provided):

Please address the remaining concern by the reviewer.

Reviewers' comments:

Reviewer's Responses to Questions

**Comments to the Author**

1. If the authors have adequately addressed your comments raised in a previous round of review and you feel that this manuscript is now acceptable for publication, you may indicate that here to bypass the “Comments to the Author” section, enter your conflict of interest statement in the “Confidential to Editor” section, and submit your "Accept" recommendation.

Reviewer #1: (No Response)

2. Is the manuscript technically sound, and do the data support the conclusions?

Reviewer #1: Yes

3. Has the statistical analysis been performed appropriately and rigorously? 

Reviewer #1: I Don't Know

4. Have the authors made all data underlying the findings in their manuscript fully available?

Reviewer #1: Yes

5. Is the manuscript presented in an intelligible fashion and written in standard English?

Reviewer #1: Yes

6. Review Comments to the Author

Reviewer #1: I would like to thank the authors on addressing the reviewer´s comments in the first round of review.

I just have one additional comment regarding the last version of the manuscript:

Regarding Table 4, and excluding the new additions regarding Visual identification, I have noticed that numbers regarding remaining categories have changed with respect to the previous version of the manuscript, and also with the parts of Table 4 included in the response to the reviewers.

Please review it carefully, and if applies, explain the reasons behind these changes and modify the corresponding parts in the manuscript text accordingly.

7. PLOS authors have the option to publish the peer review history of their article (what does this mean?). If published, this will include your full peer review and any attached files.

Reviewer #1: No

---

## [Author Response · Author response to Decision Letter 1]

8 Jan 2021

We thank the Editor and the Reviewer again for their time and effort spent reviewing our manuscript and providing us with additional suggestions and comments. We hope that our responses address the Reviewer's comments satisfactorily. 

*Please note that page numbers reflect those on the redlined document rather than the clean document file.

Reviewer #1: Regarding Table 4, and excluding the new additions regarding Visual identification, I have noticed that numbers regarding remaining categories have changed with respect to the previous version of the manuscript, and also with the parts of Table 4 included in the response to the reviewers.

Please review it carefully, and if applies, explain the reasons behind these changes and modify the corresponding parts in the manuscript text accordingly.

Response: Thank you very much for pointing out this discrepancy. We apologize for the lack of transparency.

The visual identification of bedrest was performed using a labeling software (available at https://github.com/shi-xin/actigraph_labeler), which enabled the user to discard data at the beginning and end of each Actigraph file. The previous Table 4 was produced using data files with only epochs marked as non-wearing removed. The current Table 4 was produced using data files with both non-wearing and discarded epochs removed so that visual identification and DT could be compared on the same epochs (minutes). These discarded epochs were predominantly marked as bedrest by the DT. Therefore, the removal of these epochs from analysis reduced the average bedrest (min/day) to that seen in the new Table 4. As recommended by the Reviewer, the corresponding parts in the manuscript text were modified.

Lines 144-145: The software enabled the raters to discard data at the beginning and end of each Actigraph file.

Lines 290-292: Bedrest characteristics were performed after epochs (minutes) of not-wearing and discarded in visual identification were removed.

 Lines 352-356: As identified by DT and shown in Table 4, the average daily bedrest time was over 9 hours (502 ± 62 min/day), ranging from almost 5 hours (294 min/day) to over 10 hours (652 min/day). Mean sleep duration obtained from the parental survey was significantly longer than bedrest identified by DT (650 and 502 min/day; P<0.001), Sadeh’s algorithm (650 and 604 min/day; P<0.001), and visual identification (650 and 502 min/day; P<0.001). 

Lines 371-376: On average, the wake time identified by DT was over 11 hours (734 min/day) and was longer than Sadeh's algorithm (693 min/day). The number of children taking naps reported in the parental survey (n=45) was lower than the DT identified (n=134). Naps were identified by DT in 67% of the children at least once during the monitoring period. In children whose parents reported taking daytime naps daily (23%), the naps frequency was higher than those identified by DT (1.05 ± 0.27 and 0.25 ± 0.24).

---

## [Editor Report · Decision Letter 2]

13 Jan 2021

Identifying bedrest using waist-worn triaxial accelerometers in preschool children

PONE-D-20-27484R2

Dear Dr. Buchowski,

We’re pleased to inform you that your manuscript has been judged scientifically suitable for publication and will be formally accepted for publication once it meets all outstanding technical requirements.

Kind regards,

Raffaele Ferri, MD

Academic Editor

PLOS ONE
---

## [Editor Report · Acceptance letter]

19 Jan 2021

PONE-D-20-27484R2 

Identifying bedrest using waist-worn triaxial accelerometers in preschool children 

Dear Dr. Buchowski:

I'm pleased to inform you that your manuscript has been deemed suitable for publication in PLOS ONE. Congratulations! Your manuscript is now with our production department. 

Kind regards, 

on behalf of

Dr. Raffaele Ferri 

Academic Editor

PLOS ONE